# Diagnostic accuracy of the WHO tuberculosis treatment decision algorithms for children with presumptive tuberculosis: An individual participant data meta-analysis

Laura Olbrich[1,2,3‡], Leyla Larsson[1‡*], Rory Dunbar[4], Peter J. Dodd[5], Megan Palmer[4], Minh Huyen Ton Nu Nguyet[6], Marc d'Elbée[6], Anneke C. Hesseling[4], Norbert Heinrich[1,2,3], Heather J. Zar[7], Nyanda E. Ntinginya[8], Celso Khosa[9,10], Marriott Nliwasa[11], Valsan P. Verghese[12], Maryline Bonnet[13], Eric Wobudeya[14], Bwendo Nduna[15], Raoul Moh[16], Juliet Mwanga-Amumpere[17], Ayeshatu Mustapha[18], Guillaume Breton[19], Jean-Voisin Taguebue[20], Laurence Borand[21,22], Chishala Chabala[23], Olivier Marcy[6], James A. Seddon[4,24‡], Marieke M. van der Zalm[4‡*], on behalf of the Decide-TB Study Group, the RaPaed-TB Consortium, the Umoya Study Group, and the TB Speed Consortium[¶]

**1** Institute of Infectious Diseases and Tropical Medicine, LMU University Hospital, LMU Munich, Munich, Germany, **2** German Centre for Infection Research (DZIF), Partner Site Munich, Munich, Germany, **3** Fraunhofer Institute ITMP, Immunology, Infection and Pandemic Research, Munich, Germany, **4** Desmond Tutu TB Centre, Department of Paediatrics and Child Health, Faculty of Medicine and Health Sciences, Stellenbosch University, Cape Town, South Africa, **5** Sheffield Centre for Health and Related Research, School of Medicine and Population Health, Sheffield, United Kingdom, **6** Bordeaux Population Health, Inserm U1219, IRD EMR271, University of Bordeaux, Bordeaux, France, **7** Department of Paediatrics & Child Health, Red Cross War Memorial Children's Hospital, MRC Unit on Child and Adolescent Health, University of Cape Town, Cape Town, South Africa, **8** NIMR - Mbeya Medical Research Centre, Mbeya, Tanzania, **9** Instituto Nacional de Saúde (INS), Maputo, Mozambique, **10** Department of Physiological Science, Clinical Pharmacology, Faculty of Medicine, Eduardo Mondlane University, Maputo, Mozambique, **11** Kamuzu University of Health Sciences, Blantyre, Malawi, **12** Pediatric Infectious Diseases, Christian Medical College, Vellore, India, **13** TransVIHMI, University of Montpellier, Institut de Recherche pour le Développement (IRD), Institut National de la Santé et Recherche Médicale (INSERM), Montpellier, France, **14** MU-JHU Care Ltd, Kampala, Uganda, **15** Arthur Davidson Children's Hospital, Ndola, Zambia, **16** Programme PAC-CI, Abidjan, Côte d'Ivoire, **17** Epicentre Mbarara Research Centre, Mbarara, Uganda, **18** Ola During Children Hospital, Freetown, Sierra Leone, **19** Solthis, Paris, France, **20** Mother and Child Center-Chantal Biya Foundation, Yaoundé, Cameroon, **21** Epidemiology and Public Health Unit, Institut Pasteur du Cambodge, Phnom Penh, Cambodia, **22** Center for Tuberculosis Research, Division of Infectious Diseases, Johns Hopkins University School of Medicine, Baltimore, Maryland, United States of America, **23** University of Zambia School of Medicine, University Teaching Hospital, Lusaka, Zambia, **24** Department of Infectious Disease, Imperial College London, London, United Kingdom

¶ Membership is provided in the Acknowledgments section.
‡ LO and LL share first authorship on this work. JAS and MMZ share last authorship on this work.
* Leyla.larsson@med.uni-muenchen.de (LL); mariekevdzalm@sun.ac.za (MMZ)

## Abstract

### Introduction

In 2023, almost 200,000 children under 15 years died from tuberculosis, most without appropriate treatment. Treatment decision algorithms (TDAs), developed to facilitate rapid anti-tuberculosis treatment initiation in children, were recommended by

**Data availability statement:** The IPD will be made available upon reasonable request with investigator support and a signed data access agreement. Submit request to Dr Gupta, data manager and custodian of IPD at LMU Munich (akshita.gupta@med.uni-muenchen.de). The code for analysis is available https://github.com/llarsson3/Decide-TB.

**Funding:** All authors received grant funding for this work from the third European and Developing Countries Clinical Trials Partnership programme (supported by the EU, Decide-TB, EDCTP101103283). L.O. was financially supported by a European Society for Paediatric Infectious Diseases fellowship award and a Clinical Leave Stipend from the German Center for Infection Research. M.M.Z. is supported by a career development grant from the EDCTP2 program supported by the European Union (TMA2019SFP-2836 tuberculosis lung-FACT2), the Fogarty International Centre of the National Institutes of Health (NIH) under Award Number K43TW011028, and a researcher-initiated grant from the South African Medical Research Council. The funders had no role in the study design, data collection and analysis, decision to publish, or preparation of the manuscript.

**Competing interests:** I have read the journal's policy and the authors of this manuscript have the following competing interests: N.H. has received funding from Beckman Coulter for evaluation of a test, to his institution.

**Abbreviations:** CLHIV, children living with HIV; CRS, composite reference standard; IPD, individual participant dataset; IQR, interquartile ranges; mWRD, Molecular WHO-recommended rapid diagnostic test; SD, standard deviations; TDAs, treatment decision algorithms; WHO, World Health Organization.

the World Health Organization (WHO) in 2022, conditional on validation in different cohorts and settings. We performed a retrospective external evaluation of WHO TDAs using an individual participant dataset (IPD).

## Methods and findings

The IPD comprised four paediatric cohorts, restricted to children with presumptive pulmonary TB < 10 years, and including children in high-risk groups (children living with HIV "CLHIV", children with severe acute malnutrition "SAM", and children <2 years). All children in the IPD were retrospectively evaluated using both TDA A (an algorithm including chest X-ray) and TDA B (without chest X-ray), excluding the triage step. The diagnostic accuracy against a composite reference standard (confirmed and unconfirmed tuberculosis versus unlikely tuberculosis) was determined and reported as sensitivities and specificities. Of 1,886 children included (RaPaed-TB: $n = 740$, Umoya: $n = 474$, TB-Speed HIV: $n = 204$, TB-Speed Decentralisation: $n = 468$), the median age was 2.9 years (interquartile range [IQR]:1.3,5.5), 741 (39.3%) were <2 years, 382 (20.3%) were CLHIV, and 284 (15.1%) had SAM. 281 (14.9%) had confirmed tuberculosis, 672 (35.6%) were classified as unconfirmed tuberculosis (clinically diagnosed, microbiological investigations negative), and 933 (49.5%) as unlikely tuberculosis. For TDAs A and B, algorithm sensitivity was 84.3% (95% CI: 74.8, 90.6) and 90.6% (95% CI: 83.8, 94.7), respectively, with a specificity of 50.6% (95% CI: 30.4, 70.7) and 30.8% (95% CI: 21.5, 42.0), respectively. For TDA A, estimated sensitivity in children in high-risk groups was lower than those with low-risk (83.0%, 95% CI: 79.4%, 86.1%; versus 88.0%, 95% CI: 84.8%, 90.6%), while having a gain in specificity (50.0%, 95% CI: 44.9%, 55.1%; versus 36.6%, 95% CI: 32.7%, 40.7%). Trends were similar for TDA B. As for limitations, most diagnostic tuberculosis studies in children, including two of those included in the IPD, are performed at secondary or tertiary hospitals with higher levels of healthcare and thus the target population might differ somewhat from the IPD, potentially limiting the generalisability of our results.

## Conclusions

This retrospective external evaluation of WHO TDAs in a large IPD shows high sensitivity but sub-optimal specificity for both TDAs, in line with the meta-analyses that generated the algorithms. Prospective studies that evaluate the entire TDA, including triage step are needed. Additionally, the integration of novel diagnostic tools within the TDAs should aim to enhance the accuracy, especially the specificity.

## Author summary

### Why was this study done?

- Tuberculosis in children remains one of the top 10 causes of death in those younger than 5 years, mainly due to missed or delayed diagnosis. This is

especially challenging in primary healthcare settings, where available tests are difficult to perform, require substantial infrastructure, and lack sufficient accuracy.

- In 2022, WHO recommended treatment decision algorithms for TB, which are simple flow-charts designed to guide healthcare workers step by step through a standardised diagnostic process that relies primarily on clinical information. These algorithms aim to support and standardise treatment decision, but evidence on their performance remains limited.

## What did the research find?

- In our study, we used data from several large previously conducted studies on children that underwent testing for tuberculosis, to evaluate how well these treatment decision algorithms perform to identify children with tuberculosis.

- We found that the performance in this independent dataset of children was comparable to that reported in the original discovery study. While the algorithms identified a large number of children with tuberculosis (high sensitivity), it also recommended to start a considerable number of children without tuberculosis on treatment (sub-optimal specificity).

- The accuracy was also similar in those children of vulnerable populations, including young children and those affected by HIV or malnutrition.

## What do the findings mean?

- To our knowledge, this is the first study to use previously collected data from several studies with individual participant datasets to assess the accuracy of WHO treatment decision algorithms. We validate the estimated performance using real-world data, importantly confirming its accuracy in vulnerable populations. However, low specificity might lead to substantial overtreatment, underscoring the urgent need for novel diagnostic tools with higher specificity.

- Our findings underscore the potential usefulness of diagnostic approaches such as treatment decision algorithms to identify more children eligible for tuberculosis treatment. By using a tool that can potentially be implemented at low levels of healthcare, this approach might help to avert many deaths due to childhood tuberculosis.

- Limitations include the heterogeneity of studies, partially conducted at higher levels of care, which may limit applicability to broader populations. In addition, due to the retrospective nature of the study, the initial triage/ screening step couldn't be assessed.

## Introduction

It is estimated that 1.25 million children (<15 years) develop tuberculosis annually, representing 12% of all individuals with tuberculosis globally each year [1]. In 2023, 15% of the global tuberculosis-related mortality was attributed to children [2], a disproportionate percentage compared to the incidence, despite the high efficacy of tuberculosis treatment in this age group [2]. This discrepancy is likely due to the fact that many children with tuberculosis, especially those under 5 years of age, go undiagnosed and untreated [2]. Missed diagnoses are partly due to the difficulty in confirming tuberculosis disease in children microbiologically, influenced by the paucibacillary nature of the disease, low sensitivities of available diagnostic tools, and challenges related to high-quality sample collection [3,4]. These issues are particularly relevant in primary or district levels of care, where adequate resources are not always present but where most children with tuberculosis first present [5]. In the absence of widely available accurate diagnostic tools, pragmatic approaches such as scoring systems and algorithms can assist with rapid tuberculosis treatment initiation. Treatment decision algorithms (TDAs) that generate a clinical risk score derived from the presence of symptoms or relevant medical history aim to standardise clinical decision-making and empower healthcare workers at lower levels of care.

Based on a meta-analysis including 4,718 children from 13 studies from 12 countries, two TDAs were derived using prediction modelling. One scoring system was derived using clinical and radiological features with a combined sensitivity of 0.86 [95% CI 0.68, 0.94] and specificity of 0.37 [0.15, 0.66] against a composite reference standard. A second scoring system used only clinical features and had a combined sensitivity of 0.84 [95% CI 0.66, 0.93] and specificity of 0.30 [0.13, 0.56] against a composite reference standard [6]. Following the generation of these and other evidence-based TDAs [6–9], the World Health Organization (WHO) issued a conditional recommendation for the use of TDAs in children under 10 years with presumptive pulmonary tuberculosis and included two exemplar TDAs adapted from the above-mentioned meta-analysis in their 2022 operational handbook [3,10]. One of these algorithms (TDA A) is intended for use in settings with chest X-ray (CXR), while the other (TDA B) is adapted to settings without CXR [6]. This conditional recommendation was valid for 2 years and WHO has issued a call for evidence of external validation [3]. In this study, we generate an individual participant dataset (IPD) derived from existing well-characterised paediatric TB cohorts to perform a retrospective external evaluation of the diagnostic accuracy of WHO TDAs, excluding the triage step.

## Methods

### Study design

Within the Decide-TB project, we pooled individual participant data from four large childhood TB diagnostic studies, including data from 11 countries. All studies recruited children with presumptive tuberculosis, who were prospectively recruited, and had a standardised reference tuberculosis classification applied [11]. None of these studies contributed data to the IPD used to generate WHO TDAs [10].

### Cohort-level description

Presumptive tuberculosis was defined similarly across the studies (Table A in S1 File) [12–16]. RaPaed-TB was a prospective diagnostic accuracy cohort study that recruited children <15 years with signs or symptoms of pulmonary or extrapulmonary tuberculosis in five countries (South Africa, Mozambique, Tanzania, Malawi, and India) [17]. Children were recruited in either tertiary-level hospitals or urban comprehensive healthcare facilities. Umoya was a prospective longitudinal cohort study evaluating novel diagnostic tools in South African children with presumptive pulmonary tuberculosis [13]. Recruitment was based in two secondary and tertiary-level hospitals in Cape Town. TB-Speed HIV was a prospective management study evaluating the safety and feasibility of the PAANTHER-TB TDA for hospitalised children living with HIV (CLHIV) and presumptive tuberculosis, conducted in tertiary-level healthcare centres in four countries with high-tuberculosis incidence (Côte d'Ivoire, Uganda, Mozambique, and Zambia) [14]. Lastly, TB-Speed Decentralisation was an operational research study to assess the impact of decentralising an innovative paediatric tuberculosis diagnostic approach for children with presumptive tuberculosis with a cross-sectional and nested cohort (including all children diagnosed with TB and 10% of nontuberculosis diagnosed children) design to compare two different decentralisation strategies at secondary and primary healthcare levels (22.6% primary and 77.4% district-level) [15,16]. This IPD was generated using data from children recruited to RaPaed-TB, Umoya, TB-Speed HIV, and TB-Speed Decentralisation (nested cohort only), with representation of specific groups (CLHIV, children with severe acute malnutrition "SAM", and children <2 years) identified as high-risk groups requiring immediate tuberculosis diagnostic assessment when presumptive pulmonary tuberculosis is identified (Table A in S1 File).

### Establishment of an IPD

We only included children with presumptive pulmonary tuberculosis (i.e., excluding those diagnosed by local investigators with sole extrapulmonary tuberculosis) who were <10 years, to comply with the intended use-case scenarios of both TDAs [10]. The individual datasets were prepared by extracting variables needed to recreate the TDAs in the IPD (e.g.,

presence of symptoms, tuberculosis-exposure) and conducting subsequent standardisation before merging into a single dataset, to ensure that standard scales and definitions were used [18], including the creation of composite variables (Table B in S1 File). A sensitivity analysis was conducted to assess the impact of missingness in data (Supplementary Materials).

## Treatment decision algorithms

Both WHO-TDAs include a diagnostic part that comprises microbiological testing using Xpert MTB/RIF or Ultra (Cepheid, Sunnyvale, USA) on respiratory samples, contact history, and scoring of symptoms and signs based on duration and/or presence, and an additional CXR feature scoring for TDA A (Fig A in S1 File). In both TDAs, treatment initiation is recommended if the composite score is >10. Both TDAs also include a triage step that was proposed based on expert opinion with the aim of increasing the pre-test probability of tuberculosis in the cohort (Fig A in S1 File). The triage step identifies children with low risk of disease progression (i.e., >2 years of age, not living with HIV, and not being SAM), who are recommended to be treated for most likely non-tuberculosis condition and followed up in 1–2 weeks. Considering that this is a retrospective evaluation for the assessment of TDA performance in this study, we assumed all children progressed through the algorithm to the scoring part, i.e., excluding the triage step.

**Outcome.** The diagnostic categorisation or primary outcome was defined following the updated clinical case definition for classification of intrathoracic tuberculosis in children (NIH case definition) [11]. While we primarily used the definition of the diagnostic categorisation of the primary studies, these were aligned as much as possible. In short, children were classified as having confirmed, unconfirmed, or unlikely tuberculosis following the criteria listed in Table C in S1 File [11]. Microbiological testing was performed using Xpert MTB/RIF or Ultra on respiratory samples, including spontaneous and induced sputum, nasopharyngeal aspirate, and gastric aspirate; stool results were not considered. Urine LAM was not considered as it was not standard-of-care during the original study conduct. The clinical case definitions were used to derive a composite reference standard (CRS) for diagnostic accuracy estimation, whereby children with confirmed and unconfirmed tuberculosis were defined as CRS positive and those with unlikely tuberculosis as CRS negative. In addition, we assessed the concordance between the decision-to-treat (clinical decision, CD) taken by the local clinician or healthcare worker in the four studies and the TDA recommendation for treatment initiation.

## Statistical analysis

All data management and analysis were conducted using R version 4.4.0. Cohort socio-demographic characteristics were summarised using proportions for categorical variables and medians with interquartile ranges (IQR) or means with standard deviations (SD) for continuous variables. Chi-squared testing was used to assess the difference in cohort characteristics between the included studies. The "flow" of children through the algorithm ("TDA cascade") was described, detailing who is recommended treatment initiation due to microbiological findings, presence of a tuberculosis-exposure, or a score >10. The score distribution among those eligible for scoring (i.e., no microbiological confirmation via respiratory sample and no recent tuberculosis-exposure) was described, in all eligible children and by feature of interest (e.g., HIV status, age, nutritional status, and NIH case definition). The overlap between study treatment initiation (CD) and recommendation for treatment initiation by the TDAs was described.

**Diagnostic accuracy.** Diagnostic accuracy was assessed using CRS. Firstly, TDA decision to initiate treatment was assessed and reported as sensitivities and specificities, and positive and negative predictive values (PPV, NPV) with associated 95% confidence intervals (95% CI). A McNemar's test was conducted to compare the sensitivities and specificities of the two TDAs. To account for heterogeneity between studies, a random-effects meta-analysis was conducted for the pooled estimates (R package *mada* [*reitsma* function]). This analysis was repeated for predefined subgroups of interest (risk group as defined by WHO [<2 years, SAM, or CLHIV], age, HIV status, and nutritional status [SAM versus non-SAM]).

An additional sensitivity analysis aimed to model the triage step in the beginning of the TDA. Here, we wanted to explore the impact of having a large population of children with unlikely tuberculosis that would not present to healthcare again after initial evaluation. Due to scarcity of data on tuberculosis prevalence and other diagnoses of children presenting to primary and secondary care, we randomly removed 80% of those who were considered low risk and ended up with an "unlikely tuberculosis" diagnosis.

## Ethics

All original studies obtained both ethical approval as well as documented individual consent from caretakers (and assent from children, if applicable) prior to any study-specific procedures.

## Results

A total of 2,383 children were included in the original studies (RaPaed-TB: $n = 975$, Umoya: $n = 547$, TB-Speed HIV: $n = 277$, TB-Speed Decentralisation: $n = 584$), of which 59 (2.5%) were excluded because of having extrapulmonary disease only, 91 (3.8%) excluded due to lack of diagnostic case classification (categorised as unclassifiable [Table C in S1 File] or healthy control), and finally 346 (14.5%) were excluded as older than 10 years (1 missing age), resulting in 1,886 children included in the IPD. Of these, 308 (16.3%) across all studies (RaPaed-TB: 103, UMOYA: 86, TB-Speed HIV: 12, and TB-Speed Decentralisation: 107) were missing chest X-rays.

The median age was 2.9 years (IQR: 1.3, 5.5), 741 (39.3%) were <2 years, 905 (48.0%) were female, 382 (20.3%) were CLHIV, of which 209/382 (54.7%) were already on ART and 104/382 (27.2%) were ART naive (no ART information was available from TB-Speed Decentralisation; 29/468 CLHIV), and 295 (15.9%) had SAM. 281 (14.9%) children had confirmed tuberculosis, 672 (35.6%) were classified as unconfirmed tuberculosis, and 933 (49.5%) as unlikely TB (Table 1).

### TDA cascade

Among the 1,886 children included in the TDA, we assumed that all children progress beyond the initial triage step, with those at low risk for disease progression coming back for scheduled follow-up. In total, 1,831 (97.1%) had a molecular WHO-recommended rapid diagnostic (mWRD) test result (Xpert MTB/RIF) performed on respiratory samples. Of those, 202 (11.0%) were positive and thus immediately eligible for treatment initiation (Fig 1). Of the 1,684 who did not have a positive molecular microbiological test result (negative or not done), 814 (48.3%) reported having close or household tuberculosis exposure (no time restriction), so they were recommended treatment initiation, leaving 870 children eligible for the scored section of either TDA A or B. In total 400/870 (46.0%) and 557/870 (64.0%) of the children scored >10 for TDA A and B, respectively, and thus treatment initiation would have been recommended. Study-specific cascades can be found under Figs B (TDA A) and C (TDA B) in S1 File.

### Diagnostic accuracy

The diagnostic accuracy of the TDAs against the CRS resulted in estimated pooled sensitivities of 84.3% (95% CI: 74.8, 90.6) and 90.6% (95% CI: 83.8, 94.7), and pooled specificities of 50.6% (95% CI: 30.4, 70.7) and 30.8% (95% CI: 21.5, 42.0) for TDA A and B, respectively (Table 2). Pooled PPV and NPV were 63.0% and 75.4% for TDA A and 57.2% and 76.2% for TDA B. McNemar's test resulted in statistically significant differences in terms of both sensitivity and specificity ($p < 0.001$).

There was between-study heterogeneity in estimates for both TDAs, particularly in specificities, with estimates ranging from 25.5% (95% CI: 21.4, 30.2) (RaPaed-TB, TDA A) to 73.2% (95% CI: 62.6, 81.5) (TB-Speed HIV, TDA A). Of note, as the only site recruiting at DH/PHC level, TB-Speed Decentralisation had the highest sensitivity (90.0%, 95% CI: 85.6, 93.1) and second-highest specificity (55.5%, 95% CI: 48.9, 62.0). Similar trends were observed for TDA B. Both TDAs had

**Table 1. Socio-demographic and clinical characteristics of children included in the IPD by study.**

| | Total (n = 1,886) | TB-Speed Decentralisation (n = 468) | TB-Speed HIV (n = 204) | RaPaed-TB (n = 740) | UMOYA (n = 474) | p§ |
|---|---|---|---|---|---|---|
| *Socio-demographic* | | | | | | |
| Age, years | 2.9 (1.3, 5.5) | 3.0 (1.3, 6.0) | 3.1 (1.7, 6.0) | 3.3 (1.5, 6.1) | 2.0 (0.9, 3.9) | <0.001 |
| **Age (categories)** | | | | | | <0.001 |
| <2 years | 741 (39.3%) | 172 (36.8%) | 71 (34.8%) | 260 (35.1%) | 238 (50.2%) | |
| 2 - <5 years | 596 (31.6%) | 151 (32.3%) | 64 (31.4%) | 217 (29.3%) | 164 (34.6%) | |
| 5 - <10 years | 549 (29.1%) | 145 (31.0%) | 69 (33.8%) | 263 (35.5%) | 72 (15.2%) | |
| **Sex** | | | | | | 0.52 |
| Female | 905 (48.0%) | 211 (45.1%) | 98 (48.0%) | 361 (48.8%) | 235 (49.6%) | |
| Male | 981 (52.0%) | 257 (54.9%) | 106 (52.0%) | 379 (51.2%) | 239 (50.4%) | |
| **HIV status** | | | | | | <0.001 |
| HIV-negative | 1,476 (78.3%) | 434 (92.7%) | 0 (0.0%) | 608 (82.2%) | 434 (91.6%) | |
| HIV-positive | 382 (20.3%) | 29 (6.2%) | 204 (100.0%) | 109 (14.7%) | 40 (8.4%) | |
| Unknown status | 28 (1.5%) | 5 (1.1%) | 0 (0.0%) | 23 (3.1%) | 0 (0.0%) | |
| **SAM** | 284 (15.1%) | 96 (20.5%) | 93 (45.6%) | 77 (10.4%) | 18 (3.8%) | <0.001 |
| **Case definition** | | | | | | <0.001 |
| Confirmed TB | 281 (14.9%) | 48 (10.3%) | 16 (7.8%) | 137 (18.5%) | 80 (16.9%) | |
| Unconfirmed TB | 672 (35.6%) | 202 (43.2%) | 106 (52.0%) | 227 (30.7%) | 137 (28.9%) | |
| Unlikely TB | 933 (49.5%) | 218 (46.6%) | 82 (40.2%) | 376 (50.8%) | 257 (54.2%) | |
| **Facility type** | | | | | | |
| Primary healthcare | 106 (5.6%) | 106 (22.6%) | 0 (0.0%) | 0 (0.0%) | 0 (0.0%) | |
| District healthcare | 362 (19.2%) | 362 (77.4%) | 0 (0.0%) | 0 (0.0%) | 0 (0.0%) | |
| Tertiary healthcare | 1,418 (75.2%) | 0 (0.0%) | 204 (100.0%) | 740 (100.0%) | 474 (100.0%) | |
| *TB-related findings* | | | | | | |
| **Treatment initiated** | 994 (52.7%) | 252 (53.8%) | 149 (73.0%) | 362 (48.9%) | 231 (48.7%) | <0.001 |
| **TB contact** | 916 (48.6%) | 203 (43.4%) | 38 (18.6%) | 419 (56.6%) | 256 (54.0%) | <0.001 |
| *Symptoms* | | | | | | |
| Cough ≥ 2 weeks | 1,204 (63.8%) | 345 (73.7%) | 121 (59.3%) | 540 (73.0%) | 198 (41.8%) | <0.001 |
| Fever ≥ 2 weeks | 627 (33.2%) | 241 (51.5%) | 68 (33.3%) | 259 (35.0%) | 59 (12.4%) | <0.001 |
| Weight loss | 1,029 (54.6%) | 240 (51.3%) | 138 (67.6%) | 421 (56.9%) | 230 (48.5%) | <0.001 |
| Lethargy | 719 (38.1%) | 211 (45.1%) | 108 (52.9%) | 250 (33.8%) | 150 (31.6%) | <0.001 |
| Haemoptysis | 31 (1.6%) | 11 (2.4%) | 1 (0.5%) | 15 (2.0%) | 4 (0.8%) | 0.13 |
| Night sweats | 542 (28.7%) | 85 (18.2%) | 46 (22.5%) | 292 (39.5%) | 119 (25.1%) | <0.001 |
| Enlarged lymph nodes | 324 (17.2%) | 86 (18.4%) | 48 (23.5%) | 157 (21.2%) | 33 (7.0%) | <0.001 |
| Tachycardia | 204 (10.8%) | 49 (10.5%) | 54 (26.5%) | 83 (11.2%) | 18 (3.8%) | <0.001 |
| Tachypnoea | 267 (14.2%) | 102 (21.8%) | 49 (24.0%) | 63 (8.5%) | 53 (11.2%) | <0.001 |
| *Chest X-ray pathology* | | | | | | |
| Normal | 789 (41.8%) | 249 (53.2%) | 77 (37.7%) | 302 (40.8%) | 161 (34.0%) | <0.001 |
| Cavitations | 61 (3.2%) | 6 (1.3%) | 0 (0.0%) | 46 (6.2%) | 9 (1.9%) | <0.001 |
| Adenopathy | 213 (11.3%) | 13 (2.8%) | 4 (2.0%) | 108 (14.6%) | 88 (18.6%) | <0.001 |
| Opacities | 401 (21.3%) | 62 (13.2%) | 36 (17.6%) | 193 (26.1%) | 110 (23.2%) | <0.001 |
| Miliary pattern | 41 (2.2%) | 5 (1.1%) | 7 (3.4%) | 29 (3.9%) | 0 (0.0%) | <0.001 |
| Pleural effusion | 42 (2.2%) | 16 (3.4%) | 3 (1.5%) | 19 (2.6%) | 4 (0.8%) | 0.059 |

§p-value calculated through Chi-squared testing. Abbreviations: TB, tuberculosis, SAM, severe acute malnutrition.

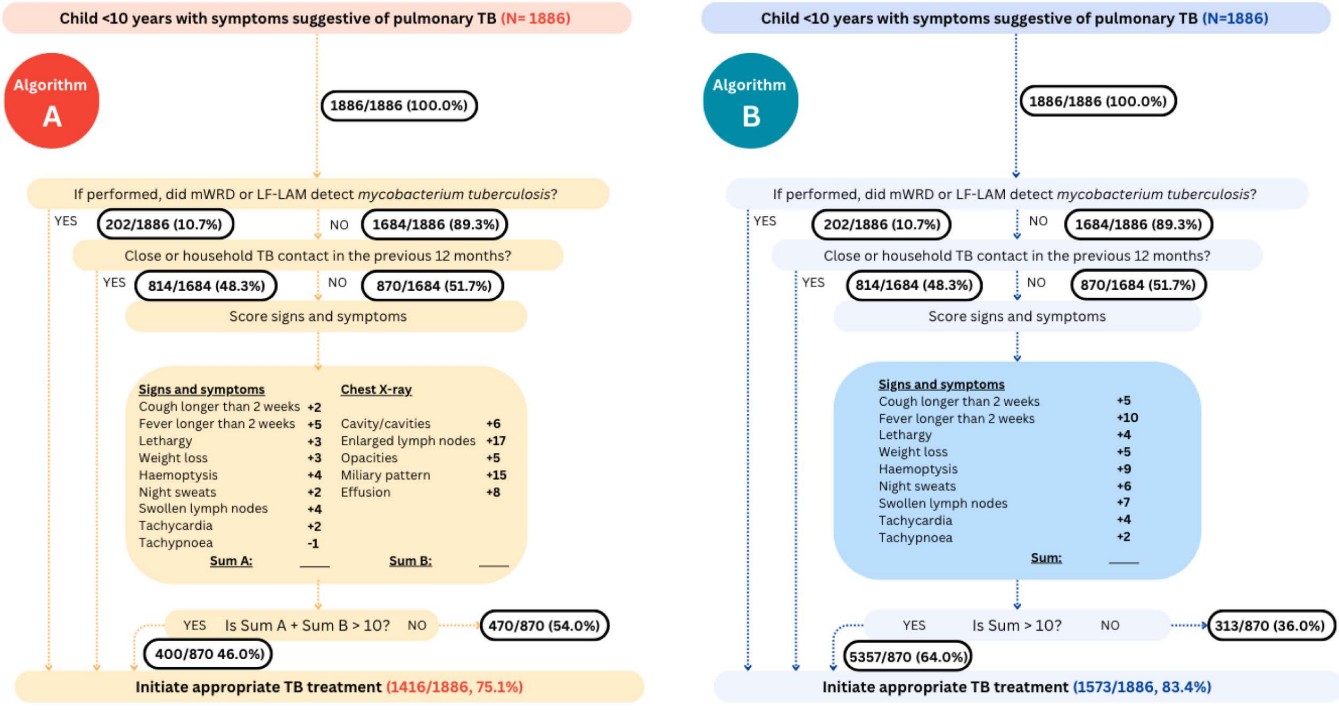

**Fig 1. Flow of children included in this study through treatment decision algorithms suggested by WHO, excluding the triage portion (figure adapted from WHO operational handbook for tuberculosis: module 5) [3].** Due to the diagnostic accuracy analysis being performed on an IPD rather than a prospective cohort, the triaging steps prior to the steps in the figure were not able to be assessed. If children were missing microbiological test results or missing findings related to presence of a TB contact, they were counted as having "NO" microbiological positivity or TB contact, reflecting how it would be implemented programmatically. Abbreviations: TDA, treatment decision algorithm; mWRD, molecular WHO-recommended rapid diagnostics; LF-LAM, lateral flow urine lipoarabinomannan; TB, tuberculosis; Sum A, sum of signs and symptoms; Sum B, sum of chest X-ray scores.

similar performance between and within the subgroups of interest, though the specificities tended to be higher in high-risk groups (<2 years, CHIV, SAM) (TDA A: high risk 50.0% [95% CI: 44.9, 55.1] versus low risk 36.6% [95% CI: 32.7, 40.7], TDA B: high risk 34.6% [95% CI: 29.9, 39.6] versus low risk 24.5% [95% CI: 21.1, 28.2]). This difference was driven mostly by increased specificity among young children (Table 2). The sensitivity analyses accounting for missingness in the data revealed no significant changes in estimates (Tables D and E in S1 File). Removing 80% of the low risk children who were classified as unlikely tuberculosis improved both specificity (TDA A: 51.1% [95% CI: 33.5, 68.3], TDA B: 33.7% [95% CI: 26.2, 42.2]) and sensitivity (TDA A: 84.3% [95% CI: 74.8, 90.6], TDA B: 90.7% [95% CI: 84.1, 94.7]) in the overall cohort.

Based on TDA A, 28.7% (542/1,886) of children would have been overtreated, i.e., falsely deemed eligible for tuberculosis treatment, especially those at low risk 34.4% (357/1,039) and those older than 2 years of age (35.9% [197/549] and 30.5% [182/596]). Undertreatment was less common with 7.3% (138/1,886) of children not being eligible for tuberculosis treatment, particularly CLHIV (14.9% [57/382]). For TDA B, patterns for overtreatment were similar with 40.9% (425/1,039) children at low risk and children older than 2 years (43.5% [239/549] and 36.7% [219/596]) being incorrectly classified as eligible. Undertreatment occurred in 4.5% (885/1,886) of children overall, particularly in CLHIV (7.3% [28/382]).

There was some concordance (85.4% TDA A, 79.9% TDA B) between the local clinician's decision-to-treat before implementation of the TDA and the retrospective TDA-based recommendation, though for both TDAs among children with unlikely tuberculosis, there was a substantial (52% TDA A, 63% TDA B) group of children who would have been started on treatment based on the TDA but for whom treatment was not initiated by the local and/or study clinicians (Fig 2).

**Table 2. Diagnostic accuracy of both treatment decision algorithms against a composite reference standard.**

| Subgroup | TP | FP | FN | TN | Sensitivity | | Specificity | |
|---|---|---|---|---|---|---|---|---|
| **TDA A** | | | | | | | | |
| **Pooled#** | 815 | 542 | 138 | 391 | **84.3%** | **(74.8%, 90.6%)** | **50.6%** | **(30.4%, 70.7%)** |
| **Study** | | | | | | | | |
| RaPaed-TB | 323 | 280 | 41 | 96 | 88.7% | (85.1%, 91.6%) | 25.5% | (21.4%, 30.2%) |
| UMOYA | 182 | 143 | 35 | 114 | 83.9% | (78.4%, 88.1%) | 44.4% | (38.4%, 50.5%) |
| TB-Speed Dec. | 225 | 97 | 25 | 121 | 90.0% | (85.6%, 93.1%) | 55.5% | (48.9%, 62.0%) |
| TB-Speed HIV | 85 | 22 | 37 | 60 | 69.7% | (61.0%, 77.1%) | 73.2% | (62.6%, 81.5%) |
| **Risk group** | | | | | | | | |
| High | 396 | 185 | 81 | 185 | 83.0% | (79.4%, 86.1%) | 50.0% | (44.9%, 55.1%) |
| Low | 419 | 357 | 57 | 206 | 88.0% | (84.8%, 90.6%) | 36.6% | (32.7%, 40.7%) |
| **HIV status** | | | | | | | | |
| HIV negative | 636 | 470 | 81 | 289 | 88.7% | (86.2%, 90.8%) | 38.1% | (34.7%, 41.6%) |
| HIV positive | 173 | 59 | 57 | 93 | 75.2% | (69.2%, 80.3%) | 61.2% | (53.2%, 68.6%) |
| Unknown | 6 | 13 | 0 | 9 | 100% | (56.8%, 93.4%) | 40.9% | (23.4%, 61.4%) |
| **Malnutrition** | | | | | | | | |
| No SAM | 668 | 484 | 112 | 338 | 85.6% | (83.0%, 87.9%) | 41.1% | (37.8%, 44.5%) |
| SAM | 147 | 58 | 26 | 53 | 85.0% | (78.9%–89.5%) | 47.7% | (38.7%–57.0%) |
| **Age** | | | | | | | | |
| <2 years | 337 | 163 | 66 | 175 | 83.6% | (79.7%, 86.9%) | 51.8% | (46.5%, 57.1%) |
| 2 -<5 years | 230 | 197 | 26 | 96 | 89.8% | (85.5%, 92.9%) | 32.8% | (27.7%, 38.3%) |
| 5 -<10 years | 248 | 182 | 46 | 120 | 84.4% | (79.7%, 88.0%) | 39.7% | (34.4%, 45.4%) |
| **TDA B** | | | | | | | | |
| **Pooled#** | 868 | 667 | 85 | 266 | **90.6%** | **(83.8%, 94.7%)** | **30.8%** | **(21.5%, 42.0%)** |
| **Study** | | | | | | | | |
| RaPaed-TB | 343 | 307 | 21 | 69 | 94.2% | (91.3%, 96.2%) | 18.4% | (14.8%, 22.6%) |
| UMOYA | 184 | 162 | 33 | 95 | 84.8% | (79.4%, 88.9%) | 37.0% | (31.3%, 43.0%) |
| TB-Speed Dec. | 237 | 149 | 13 | 69 | 94.8% | (91.3%, 96.9%) | 31.7% | (25.9%, 38.1%) |
| TB-Speed HIV | 104 | 49 | 18 | 33 | 85.2% | (77.8%, 90.4%) | 40.0% | (30.3%, 51.1%) |
| **Risk group** | | | | | | | | |
| High | 420 | 242 | 57 | 128 | 88.1% | (84.8%, 90.7%) | 34.6% | (29.9%, 39.6%) |
| Low | 448 | 425 | 28 | 138 | 94.1% | (91.6%, 95.9%) | 24.5% | (21.1%, 28.2%) |
| **HIV status** | | | | | | | | |
| HIV negative | 660 | 550 | 57 | 209 | 92.1% | (89.8%, 93.8%) | 27.5% | (24.5%, 30.8%) |
| HIV positive | 202 | 100 | 28 | 52 | 87.8% | (82.9%, 91.4%) | 34.2% | (27.2%, 42.1%) |
| Unknown | 6 | 17 | 0 | 5 | 100% | (56.8%, 93.4%) | 22.7% | (10.6%, 43.9%) |
| **Malnutrition** | | | | | | | | |
| No SAM | 712 | 579 | 68 | 243 | 91.3% | (89.1%, 93.1%) | 29.6% | (26.6%, 32.8%) |
| SAM | 156 | 88 | 17 | 23 | 90.2% | (84.8%, 93.7%) | 20.7% | (14.3%, 29.2%) |
| **Age** | | | | | | | | |
| <2 years | 355 | 209 | 48 | 129 | 88.1% | (84.6%, 90.9%) | 38.2% | (33.2%, 43.5%) |
| 2 -<5 years | 247 | 239 | 9 | 54 | 96.5% | (93.4%, 98.1%) | 18.4% | (14.4%, 23.3%) |
| 5 -<10 years | 266 | 219 | 28 | 83 | 90.5% | (86.6%, 93.3%) | 27.5% | (22.8%, 32.8%) |

Abbreviations: SAM, severe acute malnutrition; TB-Speed Dec., TB-Speed Decentralisation.

#To account for heterogeneity between studies, a random-effects meta-analysis was conducted for the pooled estimate (R package *mada* [*reitsma* function]).

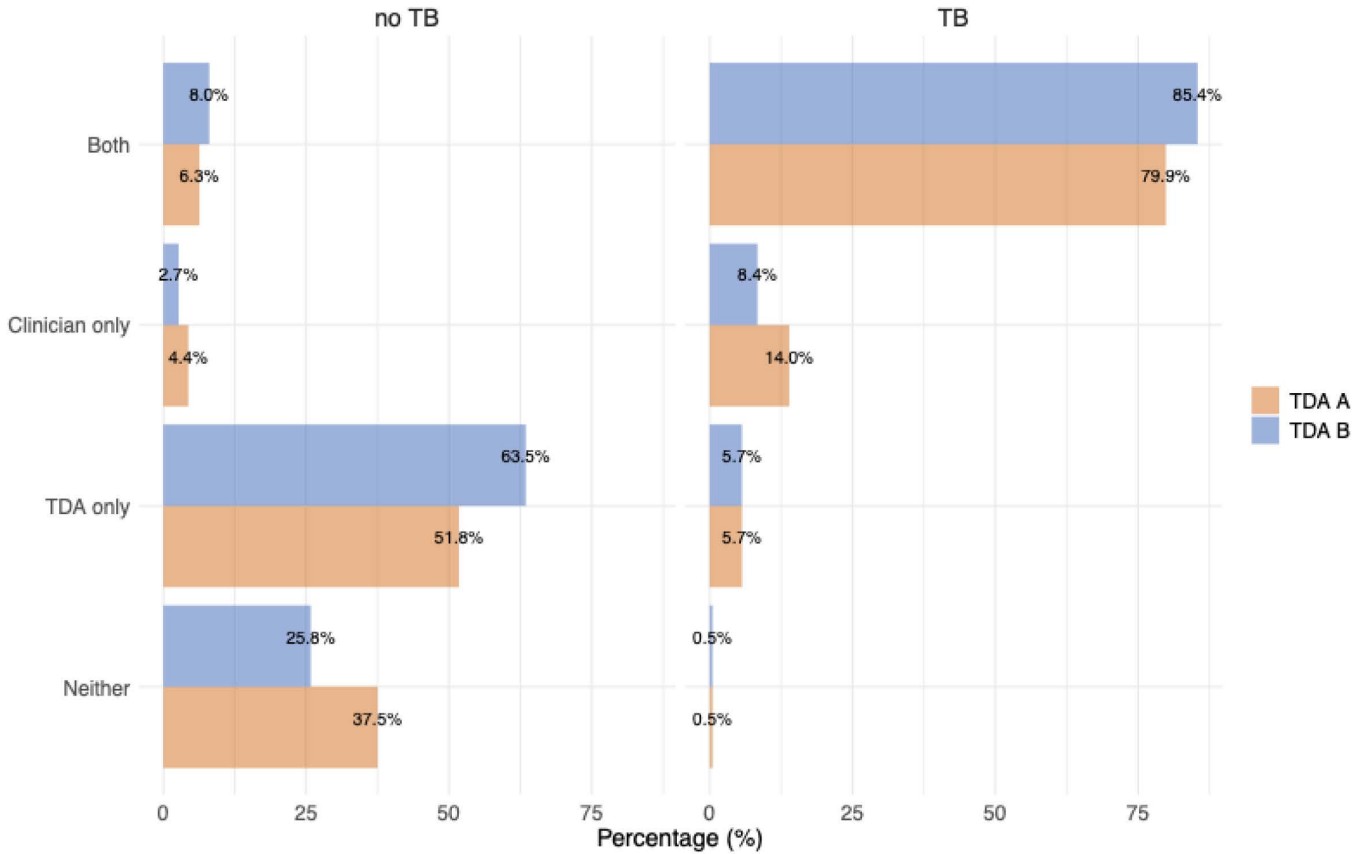

**Fig 2. Agreement between recommendation to initiate treatment according to the TDA and actual clinician decision to treat.** Proportions refer to the agreement between clinicians and the TDAs (i.e., in TDA A 85.4% of cases were in agreement between the clinician and the TDA). **A.** (Children with confirmed or unconfirmed TB and B children with unlikely TB. Agreement between TDA-recommended treatment initiation and clinician decision to treat. No treatment means neither the TDA nor the study clinicians recommended the child be initiated on treatment. Abbreviations: TDA, treatment decision algorithm.

## Score distribution

Among those who were eligible for the scoring part of the TDA (i.e., negative/no microbiological findings and no history of tuberculosis contact), children with unconfirmed tuberculosis tended to have higher median scores than children with unlikely tuberculosis (TDA A: unconfirmed 12, IQR: 7, 18; unlikely 7, IQR: 3, 11; TDA B: unconfirmed 16, IQR: 9, 24, unlikely 11, IQR: 5, 16) (Fig 3). There were no drastic differences in median scores across age in both TDAs. Both CLHIV and children with SAM reported higher median scores across all diagnostic categories (Figs 3 and D in S1 File).

## Discussion

Our external evaluation of the conditionally recommended TDAs for rapid initiation of tuberculosis treatment in children using a large retrospective IPD found that both TDAs had a high sensitivity (>84%). This is in line with the meta-analyses that generated the algorithms favouring high sensitivities to ensure children are initiated on treatment [6]. However, specificity was sub-optimal (≤50%), which would ultimately result in overtreatment for a substantial number of children and potentially missing alternative illnesses in these children. By excluding some children with a low likelihood of tuberculosis during the triage step through modelling, the specificity increased to over 50%. Future work will focus on improving the specificity

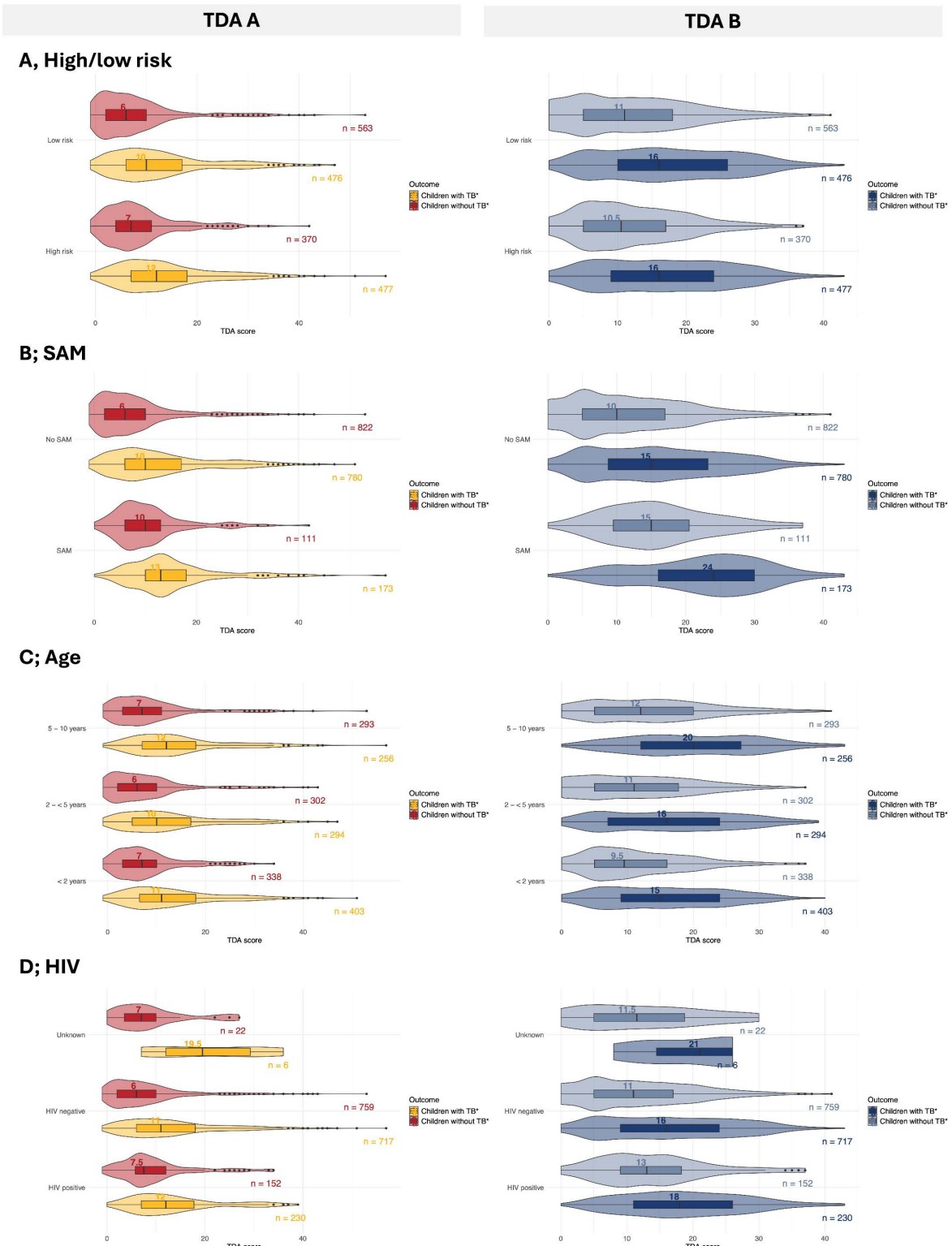

**Fig 3. Score distribution of children eligible for scoring stratified by diagnostic classification (unconfirmed [TB] vs. unlikely [no TB]). A.** High (Yes) vs. Low (No) Risk, **B.** SAM vs. no SAM, **C.** Age, **D.** HIV status. Abbreviations: SAM, severe acute malnutrition, TB, tuberculosis.

of these TDAs while maintaining their ability to identify children in need of treatment. This will include integrating novel diagnostic tools, such as biomarkers and additional artificial intelligence-based imaging techniques, to enhance the accuracy of the algorithms. Additionally, evaluating and optimising the triage step within the TDAs will be a priority to ensure that fewer children are unnecessarily treated while those at the highest risk are prioritised. These improvements aim to refine the TDAs, ultimately aiming to close the diagnostic gap and improve outcomes, especially in low-resource settings.

The studies included in this IPD represent a wide-range of settings where childhood tuberculosis diagnoses are made, including 11 different countries, with RaPaed-TB, Umoya, and the TB-Speed HIV study being rigorously conducted tuberculosis diagnostic studies implemented at higher levels of healthcare and within well-resourced research settings [12,13]. On the other hand, the TB-Speed Decentralisation study was a large study, implemented at lower levels of healthcare (primary and district-level) where nurses could be in charge of tuberculosis diagnosis, similar to where TDAs are intended for use. Despite these differences in setting, the performance of the TDAs in terms of sensitivity was similar across these studies and the sensitivity was in fact highest in primary and district-level settings. The specificity, however, varied across settings, with the highest specificity achieved in the high-risk groups of children, mostly driven by higher specificities in younger children (<2 years). The varying specificity in different studies is likely due to the diagnostic classification, as RaPaed-TB and Umoya underwent more rigorous review processes. The low overall specificity is concerning, especially in light of the imperfect reference standard, potentially leading to an overestimation in our dataset. In future, the addition of novel tools with high specificity might increase overall accuracy and especially specificity.

However, due to the generally high morbidity and mortality associated with undiagnosed and untreated tuberculosis in children, the safety profile of the treatment regimen, and the excellent outcomes for those who initiate treatment, a degree of overtreatment is generally accepted [19,20]. This tendency of some overtreatment with the TDAs was evident when comparing the TDAs to the local clinician's decision to treat. Among the children classified as unlikely tuberculosis, over 50% of children recommended for treatment by the TDAs were not treated by the local (study) clinician. Overtreatment was particularly common in children with low risk and those older than 2 years, where around a third were falsely identified as eligible for treatment. In contrast, undertreatment was low overall. It is important to note, however, that these study clinicians are generally very experienced in diagnosing tuberculosis in children and had more resources available to them, while the TDAs are largely recommended for use in primary health facilities, which are often run by staff with less tuberculosis-specific training and more limited resources [21].

The TDA without CXR (TDA B), had higher sensitivity than TDA A, but performed worse in terms of specificity. This has important programmatic implications, suggesting the need for a stronger advocacy for wide-spread availability of CXR. Considering infrastructural constraints, higher specificities could result in lower resource use and better management of nontuberculosis pathologies. However, it is important to consider that the CXR classifications included in this IPD were based on the interpretation of the CXR by tuberculosis experts, which might not reflect the CXR interpretation of less experienced readers. Furthermore, CXR can be challenging to interpret, costly to healthcare systems, and not all health workers are trained or accredited to read CXR for tuberculosis diagnosis [22]. Computer-aided detection of CXR for the diagnosis of tuberculosis could overcome this barrier [23].

It is important to highlight that almost half of the children had a TDA decision to initiate treatment, and thus excluded from the score assessment, due to the fact that they had a history of a tuberculosis-exposure. While this step in the algorithm cannot be directly evaluated, it may contribute to a higher rate of overtreatment. However, initiating treatment in this context is vital because children with a history of tuberculosis-exposure are a vulnerable group at substantial risk of progressing to tuberculosis disease [24]. Not all symptomatic children with such a history necessarily require treatment, but early intervention in high-risk children is essential to prevent disease progression and its associated complications [25].

The strengths of this study include the well-characterised and comprehensive dataset of children presenting with presumptive pulmonary tuberculosis covering a wide-range of geographical areas in regions with a high-tuberculosis burden. Most of the variables needed to model the performance of the TDAs were already collected and available in the individual

studies, limiting the need for additional assumptions and imputation. As for limitations, although the study included high-level tuberculosis diagnostic studies and a large study conducted at primary healthcare levels, the heterogeneity of the studies led to uncertainty in the findings, though this improved generalisability. During study conduct, urine LAM-testing was not standard-of-care and thus not considered here. Most diagnostic tuberculosis studies in children, including two of those included in the IPD, are performed at higher levels of healthcare and thus the target population might differ somewhat from the IPD. Despite the considerable effort and resources that went into these studies, a major inherent limitation is the uncertainty of true tuberculosis status of children without microbiological confirmation, resulting in an unclear estimation of diagnostic performance of TDAs. This also includes potential incorporation bias, for example, as there is an overlap between the reference standard definition (i.e., unconfirmed tuberculosis) and TDA scoring. In addition, because of the retrospective nature of the IPD evaluation, the "triage" steps prior to the microbiological, tuberculosis history, and scoring sections of the algorithm cannot be assessed, and these could modify overall sensitivity and specificity of the TDAs. Prospective studies are needed to see how the triage step impacts the diagnostic accuracy of these TDAs and to evaluate the feasibility and acceptability of the TDAs in practice. This also includes the potential development of similar tools for the use in active case finding among children without symptoms in future.

In conclusion, it is encouraging to see that the external validation of WHO TDAs is highly sensitive and shows robust performance across settings. However, their specificity is sub-optimal, which may hamper their acceptability. Prospective evaluations to allow assessment of the entire algorithm, including the triage step are an important next step. This also includes to explore different thresholds for the scoring section, especially in subgroups of interest, which may also be dependent on settings, symptoms, and other characteristics. It will also be important to combine these TDAs with novel diagnostic tools, including biomarkers and AI-based tools, or sampling strategies to further improve TDA specificities, improve efficiency and ensure their feasibility for use in lower levels of care.

## Supporting information

**S1 File.** Fig A: WHO treatment decision algorithms (TDAs) adapted from WHO operational handbook on tuberculosis. Module 5: management of tuberculosis in children and adolescents. Geneva: World Health Organization; 2022. Licence: CC BY-NC-SA 3.0 IGO [3]. **Table A:** Study characteristics of individual studies included in the IPD; TB-Speed decentralisation, TB-Speed HIV, RaPaed-TB and Umoya. **Table B:** Variable definitions of individual studies included in the IPD; TB-Speed decentralisation, TB-Speed HIV, RaPaed-TB and Umoya. **Table C:** Outcome definitions used in the study. **Fig B:** Cascade through TDA A stratified by included study in the IPD. For TB-Speed HIV, this is based on the imputed dataset. WHO TDA images adapted from WHO operational handbook on tuberculosis. Module 5: management of tuberculosis in children and adolescents. Geneva: World Health Organization; 2022. Licence: CC BY-NC-SA 3.0 IGO [3]. **Fig C:** Cascade through TDA B stratified by included study in the IPD. For TB-Speed HIV, this is based on the imputed dataset. WHO TDA images adapted from WHO operational handbook on tuberculosis. Module 5: management of tuberculosis in children and adolescents. Geneva: World Health Organization; 2022. Licence: CC BY-NC-SA 3.0 IGO [3]. **Table D:** Diagnostic accuracy of both treatment decision algorithms against a composite reference standard assuming all night sweats in the TB-Speed HIV cohort are absent. **Table E:** Diagnostic accuracy of both treatment decision algorithms against a composite reference standard assuming all night sweats in the TB-Speed HIV cohort are present. **Fig D:** Score distribution of children eligible for scoring stratified by site **A.** TDA A, **B.** TDA B.
(DOCX)

## Acknowledgments

We would like to acknowledge all individual study groups and participants as well as the scientific advisory board of the Decide-TB project: Elizabeth Maleche Obimbo, Stephen M. Graham, Moorine P. Sekkade, Andrew Copas, Sabine Verkuijl, Jenny Hill, Anna Scardigli, Anne K. Detjen, Albert Kuaté, Sharon Musakanya, and Charity Habeenzu.

## Author contributions

**Conceptualization:** Laura Olbrich, Leyla Larsson, James A. Seddon, Marieke M. van der Zalm.

**Data curation:** Laura Olbrich, Leyla Larsson, Rory Dunbar, Minh Huyen Ton Nu Nguyet, Marieke M. van der Zalm.

**Formal analysis:** Laura Olbrich, Leyla Larsson, Rory Dunbar, Minh Huyen Ton Nu Nguyet, James A. Seddon, Marieke M. van der Zalm.

**Funding acquisition:** Decide TB authors.

**Investigation:** Laura Olbrich, Leyla Larsson, Rory Dunbar, Peter J. Dodd, Megan Palmer, Minh Huyen Ton Nu Nguyet, Marc d'Elbée, Anneke C. Hesseling, Norbert Heinrich, Heather J. Zar, Nyanda E. Ntinginya, Celso Khosa, Marriott Nliwasa, Valsan P. Verghese, Maryline Bonnet, Eric Wobudeya, Bwendo Nduna, Raoul Moh, Juliet Mwanga-Amumpere, Ayeshatu Mustapha, Guillaume Breton, Jean-Voisin Taguebue, Laurence Borand, Chishala Chabala, Olivier Marcy, James A. Seddon, Marieke M. van der Zalm, Decide-TB Study Group, the RaPaed-TB Consortium, the Umoya Study Group, and the TB Speed Consortium.

**Methodology:** Laura Olbrich, Leyla Larsson, James A. Seddon, Olivier Marcy, Marieke M. van der Zalm.

**Project administration:** Laura Olbrich, Leyla Larsson, Olivier Marcy, Marieke M. van der Zalm.

**Supervision:** James A. Seddon, Marieke M. van der Zalm.

**Visualization:** Leyla Larsson.

**Writing – original draft:** Laura Olbrich, Leyla Larsson, Marieke M. van der Zalm.

**Writing – review & editing:** Laura Olbrich, Rory Dunbar, Peter J. Dodd, Megan Palmer, Minh Huyen Ton Nu Nguyet, Marc d'Elbée, Anneke C. Hesseling, Norbert Heinrich, Heather J. Zar, Nyanda E. Ntinginya, Celso Khosa, Marriott Nliwasa, Valsan P. Verghese, Maryline Bonnet, Eric Wobudeya, Bwendo Nduna, Raoul Moh, Juliet Mwanga-Amumpere, Ayeshatu Mustapha, Guillaume Breton, Jean-Voisin Taguebue, Laurence Borand, Chishala Chabala, Olivier Marcy, James A. Seddon.

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
