## [Editor Report · Decision Letter 0]

4 Mar 2025

Dear Dr van der Zalm,

Thank you for submitting your manuscript entitled "Diagnostic accuracy of the WHO-recommended tuberculosis treatment decision algorithms for children with presumptive tuberculosis: an individual participant data meta-analysis" for consideration by PLOS Medicine.

Your manuscript has now been evaluated by the PLOS Medicine editorial staff as well as by an academic editor with relevant expertise and I am writing to let you know that we would like to send your submission out for external peer review.

Please re-submit your manuscript within two working days, i.e. by Mar 06 2025 11:59PM.

Kind regards,

Alison Farrell, Ph.D.

Senior Editor

PLOS Medicine

---

## [Decision Letter · Decision Letter 1]

3 Jun 2025

Dear Dr van der Zalm,

Many thanks for submitting your manuscript "Diagnostic accuracy of the WHO-recommended tuberculosis treatment decision algorithms for children with presumptive tuberculosis: an individual participant data meta-analysis" (PMEDICINE-D-25-00818R1) to PLOS Medicine. The paper has been reviewed by subject experts and a statistician; their comments are included below and can also be accessed here: [LINK]

As you will see, while the referees find the study interesting, they raise concerns about the extent to which it helps move forward TB diagnostics for children with low mycobacterial load. Moreover, they indicate that the manuscript needs to more accurately frame the analysis relative to the WHO decision algorithm, increase transparency of the methodology and results, and more extensively discuss study limitations and the related literature, among other issues. After discussing the paper with the editorial team and an academic editor with relevant expertise, I'm pleased to invite you to revise the paper in response to the reviewers' comments. We plan to send the revised paper to some or all of the original reviewers, and we cannot provide any guarantees at this stage regarding publication. We may seek the assistance of a statistician to review the study.

We ask that you submit your revision by Jun 24 2025 11:59PM. However, if this deadline is not feasible, please contact me by email, and we can discuss a suitable alternative.

Don't hesitate to contact me directly with any questions (afarrell@plos.org).

Best regards,

Alison

Alison Farrell, Ph.D.

Senior Editor

PLOS Medicine

afarrell@plos.org

Comments from the academic editor:

1) "Exclusion of participants with extra-pulmonary TB from analysis". Given the challenges of diagnosing all forms of TB in young children, and the high rate of extra-pulmonary TB, I struggle to see how the authors could have reliably done this. Additionally, I note that participants with LF-LAM positive urine were included in analysis, and this would usually indicate extra-pulmonary TB - needs clarification.

2) Likewise, although mWRD positivity indicated treatment, it is not clear whether these tests were done on sputum, gastric aspirate, stool, or urine. Needs clarification, and specimen type may change case definition.

3) I didn't fully understand the analysis where 80% of "low risk" participants were removed from analysis to investigate triage testing approaches - I suspect that, given the considerable diagnostic uncertainty, this would not be a robust approach.

4) As well as sensitivity and specificity estimates, important to show pooled estimates of positive and negative predictive values.

5) Discussion is currently unreferenced, and needs greater critique of the limitations of the study.

6) The authors suggest that using AI analysis of chest X-ray could improve specificity - I think the opposite is likely to be true - chest X-ray signs are generally non-specific for TB, and I suspect that AI algorithms will struggle to distinguish TB from non-TB findings, notwithstanding the fact that diagnostic accuracy CXR studies are very difficult to do in children. Addition of another low specificity test to a low-specificity algorithm will likely further worsen over-treatment. This proposal needs much greater consideration and supporting evidence.

Comments from the reviewers:

Reviewer #1: General comments

This study retrospectively assessed the diagnostic accuracy of the 2022 WHO-recommended treatment decision algorithms for TB in children under 10 years of age using individual data from four studies. This is a well-conducted study that includes a large number of children and uses data from previous, rigorously conducted studies. The results reported are novel and important. The manuscript is clear and well-written.

My main comment is related to the fact that the results of the study inform about part of the algorithms (after what is called by the authors the "triage step") and not about the entire WHO-recommended algorithms. This is particularly relevant for the interpretation of the accuracy. Although this is mentioned in some parts of the manuscript, it is not stated in the abstract, and could be further highlighted in the introduction and the methods.

Abstract

I suggest stating in the introduction section and/or in the conclusions that the evaluation of the algorithms and the findings concern only part of the WHO-recommended algorithms (and not the entire WHO-recommended algorithms).

I suggest adding in the methods paragraph that children were "retrospectively" evaluated using both TDA A and TDA B.

I think the last sentence in the conclusion is not clear. It is stated that "prospective studies should aim to enhance the specificity of these TDAs". Should this read that "prospective studies should assess the specificity of these TDAs"?

Methods

Cohort level description: I suggest including in the description that the Umoya cohort excluded children who had an alternative diagnosis at baseline. This information is important as treating non-TB conditions is one of the steps in the WHO-recommended algorithms. I also suggest including in the supplemental table 1 the definition of presumptive TB used in each of the four studies, and if possible, a summary of these definitions in the body of the manuscript.

Treatment Decision Algorithms: This section mentions that the algorithms include a "triage step". I suggest adding what this step consists of and stating that this step was excluded from the evaluation, and that therefore, the findings do not concern the entire WHO-recommended algorithms.

Outcome: Can the authors clarify whether each participant was classified in a diagnostic category for the meta-analysis or if the diagnostic categorization by the primary studies was used?

Discussion

Paragraph strengths and limitations: minor comment, consider stating that the limitations start at the beginning of the sentence where the first limitation is mentioned.

Reviewer #2: PMEDICINE-D-25-00818R1

General comments

* The authors performed a retrospective external evaluation of WHO Treatment Decision Algorithms (TDAs) for childhood TB, using an individual participant dataset. The WHO TDAs were recommended by WHO in 2022, conditional on validation in different cohorts and settings. Development of the WHO TDAs was originally based on meta-analysis of a large individual participant dataset, using a previously published composite reference standard for Confirmed and Unconfirmed TB (Graham et al, CID 2015).

* The main findings can be summarized by the Take Home Message: "When the WHO treatment-decision algorithms for children with presumptive tuberculosis were applied to an independent retrospective individual participant dataset, the performance was similar to that on the discovery dataset, with similar performance in vulnerable populations. Specificity remained sub-optimal".

* Although the methodology was sound and the goal of the study was achieved, it is not entirely clear how the findings might be useful. Since the majority of childhood TB is not confirmed microbiologically, the study "validates" one algorithm for unconfirmed TB against another algorithm for unconfirmed TB. The findings bring us no closer to improved diagnostics for paucibacillary childhood TB, given the lack of a reliable gold standard.

Specific comments:

Abstract

* The abstract conclusion refers to high sensitivity, but suboptimal specificity of both TDAs. This is hardly surprising since the TDAs were designed around these parameters (see below). Suggest conclude that the observed performance was similar to that of the discovery dataset from which the TDAs were derived.

Introduction.

* The need for sensitive diagnostics for childhood TB is discussed, but the need for more specific diagnostics should also be noted, given the findings and limitations below.

* Suggest note in the Introduction the sensitivity and specificity of the scoring systems in the original TDA prediction models, which included CXR (86% sensitivity and 37% specificity), or only clinical features (84% sensitivity and 30% specificity), per Gunasekera et al, Lancet Adolesc Health 2023.

Methods

* The dataset is large, representative and fit for purpose.

Results

* Diagnostic accuracy of the TDAs against the composite reference standard is reported as pooled sensitivities of 84.3% and 90.6%, and pooled specificities of 50.6% and 30.8%, for TDA A (with CXR) and B (clinical features only), respectively. As noted above, this performance is similar to that of the discovery IPD analysis (Gunasekera et al).

* It would be useful to describe the characteristics of the group of false-negative TB cases that were missed, and who would not be treated by application of the TDAs; similarly, to describe the group of false-positive TB cases that would be treated unnecessarily by application of the TDAs.

* The main text Figure legends appear to missing from the file.

Discussion

* The main limitation of the study is the same as that of the original TDA discovery study, ie. uncertainty of diagnosis of unconfirmed TB included in the composite reference standard. The authors refer to "sub-optimal specificity" against this composite reference standard, but true specificity is probably lower. It would be useful to note the implications of this limitation for TDA false-positive diagnosis, unnecessary treatment burden, and treatment-related adverse effects.

* Some discussion of the group characteristics of TDA false-negative and TDA false-positive TB cases would provide useful insight into the consequences of under- and over-treatment.

* A limitation of this analysis, and of the datasets used to generate the TDAs, and of the composite reference standard, is that they include only symptomatic children presenting with presumptive pulmonary TB. Suggest note as a limitation that the findings are not generalizable to active case-finding among children without symptoms who have not presented for care.

Reviewer #3: The authors conducted a pooled analysis of four well-recognized studies in childhood TB to retrospectively calculate the accuracy of the WHO treatment decision algorithms. This is an important effort given the conditional recommendation, and can guide current use and directions for improvement including the need for higher specificity.

MAJOR

1. Abstract - would clarify that the accuracy is being measured after the triage phase/at the point of testing and clinical evaluation

2. Abstract - please add the accuracy when stratified by high vs. low risk children

3. Overall it is not clear at which point the accuracy is being measured at the testing/evaluation phase. The total sample size in the IPD analysis is 1,886, but then there is a focus on the scoring part (page 6 "in this study, we assumed all children progressed through the algorithm to the scoring part"). Is this the accuracy if all children were scored, or accuracy if the TDA was positive (i.e. +molecular testing, contact, or score >10)? Please clarify in methods/results.

4. TB exposure was not considered to be in the scoring part, but actually it is part of the scoring; in the meta-analysis (PMID: 36924781), TB contact had a score >10 and therefore was moved out of the box, and is part of the model used to calculate the other scores. Thus, would include it when considering the accuracy and score distribution.

5. At least for this reviewer, there does not appear to be figure titles or legends, and Figure 3 is mislabeled as Figure 2 in the scoring section of results.

6. Figure 2 overall is not clear, do the each bar represent proportion positive? I also assume no TB and TB are based on the CRS, but this makes the clinician only interpretation challenging as their decisions influence the definition of Unconfirmed TB.

7. "There was some concordance" (Page 12). Please quantify the overall concordance and discordance

8. Figure 3 - if excluded those with negative/no microbiological findings, then by definition this is looking at Unconfirmed vs. Unlikely TB, but the text (page 12) notes Confirmed or Unconfirmed? How can those with Confirmed TB then have a median score?

9. Figure 3 - I think we would expect higher scores given the overlap between the scoring system and having Unconfirmed TB? Given the scores are higher, is there a higher score threshold that could be then used to increase specificity without significant loss of sensitivity? The implication and limitations of these findings should be in the discussion.

10. There will be incorporation bias with the CRS as the definitions for Confirmed and Unconfirmed TB include microbiological testing, and there is overlap in the scoring and Unconfirmed definition. Please discuss this limitation and the implications in the discussion.

11. LAM testing is also part of the algorithm among children with HIV, but not mentioned, please include or note as a limitation

MINOR

1. Table 1 - could you add the proportion that has normal CXRs?

2. For the analysis where 80% of low-risk children with Unlikely TB are excluded, clarify if this accuracy is overall or just for the low-risk children

3. "In addition, the treatment of tuberculosis infection (using 2 drugs for 3 months) and non-severe disease (using 3-4 drugs for 4 months) are fairly similar which limits the additional burden to have a low threshold of initiating treatment in these children." This is an oversimplification of the differences in TB infection and disease management, and associated other challenges including missed alternative diagnoses, stigma, isolation, costs, etc. While the importance of empiric treatment is noted, would suggest still emphasizing the authors' main point that greater specificity is still needed.

---

* Please upload any figures associated with your paper as individual TIF or EPS files with 300dpi resolution at resubmission; please read our figure guidelines for more information on our requirements: http://journals.plos.org/plosmedicine/s/figures. While revising your submission, please upload your figure files to the PACE digital diagnostic tool, https://pacev2.apexcovantage.com/. PACE helps ensure that figures meet PLOS requirements. To use PACE, you must first register as a user. Then, login and navigate to the UPLOAD tab, where you will find detailed instructions on how to use the tool. If you encounter any issues or have any questions when using PACE, please email us at PLOSMedicine@plos.org.

* Please provide a detailed Data Availability Statement, explaining all restrictions to data access or use.

* Please ensure that the study is reported according to the relevant guideline and include the completed checklist as Supporting Information. When completing the checklist, please use section and paragraph numbers, rather than page numbers. Please add the following statement, or similar, to the Methods: "This study is reported as per [XXXX] guideline (S1 Checklist)."

FIGURES AND TABLES

SUPPLEMENTARY MATERIAL

REFERENCES

OBSERVATIONAL STUDIES

* Abstract: Please include the study design, population and setting, number of participants, years during which the study took place (enrollment and follow up), length of follow up, and main outcome measures.

* Please ensure that the study is reported according to the STROBE (or appropriate STOBE extension) guideline (available from: https://www.equator-network.org/reporting-guidelines/strobe) and include the completed STROBE (or STROBE extension) checklist as Supporting Information. Please add the following statement, or similar, to the Methods: "This study is reported as per the Strengthening the Reporting of Observational Studies in Epidemiology (STROBE) guideline (S1 Checklist)." When completing the checklist, please use section and paragraph numbers, rather than page numbers.

* [FOR POPULATION HEALTH/REGISTRY STUDIES] Please ensure that the study is reported according to the RECORD guideline (available from https://www.record-statement.org) and include the completed checklist as Supporting Information. Please add the following statement, or similar, to the Methods: "This study is reported as per the Reporting of Studies Conducted using Observational Routinely-Collected Data (RECORD) guideline (S1 Checklist)." When completing the checklist, please use section and paragraph numbers, rather than page numbers.

* [FOR POPULATION HEALTH ESTIMATES] Please ensure that the study is reported according to the GATHER statement (available from https://www.equator-network.org/reporting-guidelines/gather-statement) and include the completed checklist as Supporting Information. Please add the following statement, or similar, to the Methods: "This study is reported as per the Guidelines for Accurate and Transparent Health Estimates Reporting (GATHER) statement (S1 Checklist)." When completing the checklist, please use section and paragraph numbers, rather than page numbers.

* [FOR MEDIATION ANALYSES] We recommend that the study is reported according to the AGReMA statement (https://agrema-statement.org/#:~:text=AGReMA%20is%20an%20evidence%2D%20and,randomised%20trials%20and%20observational%20studies) and include the completed checklist as Supporting Information. Please add the following statement, or similar, to the Methods: "This study is reported as per the Guideline for Reporting Mediation Analyses (AGReMA) statement (S1 Checklist)." When completing the checklist, please use section and paragraph numbers, rather than page numbers.

* For all observational studies, in the manuscript text, please indicate: (1) the specific hypotheses you intended to test, (2) the analytical methods by which you planned to test them, (3) the analyses you actually performed, and (4) when reported analyses differ from those that were planned, transparent explanations for differences that affect the reliability of the study's results. If a reported analysis was performed based on an interesting but unanticipated pattern in the data, please be clear that the analysis was data driven.

* Please state in the Methods section whether the study had a prospective protocol or analysis plan. If a prospective analysis plan (from your funding proposal, IRB or other ethics committee submission, study protocol, or other planning document written before analyzing the data) was used in designing the study, please include the relevant document(s) with your revised manuscript as a Supporting Information file to be published alongside your study and cite it in the Methods section. A legend for this file should be included at the end of your manuscript. If no such document exists, please make sure that the Methods section transparently describes when analyses were planned, and when/why any data-driven changes to analyses took place. Changes in the analysis, including those made in response to peer review comments, should be identified as such in the Methods section of the paper, with rationale.

MODELLING STUDIES

The following list is derived from Geoffrey P Garnett, Simon Cousens, Timothy B Hallett, Richard Steketee, Neff Walker. Mathematical models in the evaluation of health programmes. (2011) Lancet DOI:10.1016/S0140-6736(10)61505-X:

* If pertinent, please provide a diagram that shows the model structure, including how the natural history of the disease is represented, the process and determinants of disease acquisition, and how the putative intervention could affect the system.

* Please provide a complete list of model parameters, including clear and precise descriptions of the meaning of each parameter, together with the values or ranges for each, with justification or the primary source cited and important caveats about the use of these values noted.

* Please provide a clear statement about how the model was fitted to the data, including goodness-of-fit measure, the numerical algorithm used, which parameter varied, constraints imposed on parameter values, and starting conditions.

* For uncertainty analyses, please state the sources of uncertainties quantified and not quantified [can include parameter, data, and model structure].

* Please provide sensitivity analyses to identify which parameter values are most important in the model. Uncertainty estimates seek to derive a range of credible results on the basis of an exploration of the range of reasonable parameter values. The choice of method should be presented and justified.

* Please discuss the scientific rationale for the choice of model structure and identify points where this choice could influence conclusions drawn. Please also describe the strength of the scientific basis underlying the key model assumptions.

* For studies that develop a prediction model or evaluate its performance, please ensure that the study is reported according to the TRIPOD statement (https://www.equator-network.org/reporting-guidelines/tripod-statement) and include the completed checklist as Supporting Information. Please add the following statement, or similar, to the Methods: "This study is reported as per the Transparent Reporting of a Multivariable Prediction Model for Individual Prognosis Or Diagnosis (TRIPOD) statement (S1 Checklist)." For studies using machine learning, please use the TRIPOD-AI checklist. When completing the checklist, please use section and paragraph numbers, rather than page numbers.

DIAGNOSTIC STUDIES

* Please ensure that the study is reported according to the STARD guideline (https://www.equator-network.org/reporting-guidelines/stard/) and include the completed STARD checklist as Supporting Information. Please add the following statement, or similar, to the Methods: "This study is reported as per the Standards for Reporting of Diagnostic Accuracy (STARD) guideline (S1 Checklist)." When completing the checklist, please use section and paragraph numbers, rather than page numbers.

* Please structure your Abstract according to STARD for Abstracts (https://www.equator-network.org/reporting-guidelines/stard-abstracts/).

* Please structure the Methods section using the following sub-headings: Study design, Participants, Test methods, Analysis.

* Please include a diagram to describe the flow of participants through the study (typically figure 1).

QUALITATIVE STUDIES

* Please report your qualitative study according to the appropriate study design provided at (http://www.equator-network.org/?post_type=eq_guidelines&eq_guidelines_study_design=qualitative-research&eq_guidelines_clinical_specialty=0&eq_guidelines_report_section=0&s=) and provide the relevant completed checklist as a supplemental file. In the checklist, please include sufficient text excerpted from the manuscript to explain how you accomplished all applicable items. When completing checklists, please use section and paragraph numbers, rather than page numbers.

* We recommend that authors use the COREQ checklist, or other relevant checklists listed by the Equator Network, such as the SRQR, to ensure complete reporting (see: http://www.equator-network.org/?post_type=eq_guidelines&eq_guidelines_study_design=qualitative-research&eq_guidelines_clinical_specialty=0&eq_guidelines_report_section=0&s=). Please add the following statement, or similar, to the Methods: "This study is reported as per the Consolidated criteria for reporting qualitative research (COREQ): a 32-item checklist for interviews and focus groups (S1 Checklist)."

* In general, we expect qualitative studies to include the following: 1) defined objectives or research questions; 2) description of the sampling strategy, including rationale for the recruitment method, participant inclusion/exclusion criteria and the number of participants recruited; 3) detailed reporting of the data collection procedures; 4) data analysis procedures described in sufficient detail to enable replication; 5) a discussion of potential sources of bias; and 6) a discussion of limitations.

---

## [Decision Letter · Decision Letter 2]

9 Sep 2025

Dear Dr. van der Zalm,

Thank you very much for re-submitting your manuscript "Diagnostic accuracy of the WHO tuberculosis treatment decision algorithms for children with presumptive tuberculosis: an individual participant data meta-analysis" (PMEDICINE-D-25-00818R2) for review by PLOS Medicine.

I have discussed the paper with my colleagues and the academic editor and it was also seen again by 3 reviewers. I am pleased to say that provided the remaining editorial and production issues are dealt with we are planning to accept the paper for publication in the journal.

The remaining issues that need to be addressed are listed at the end of this email. Any accompanying reviewer attachments can be seen via the link below.

Please ensure that you address in the manuscript the remaining minor comments from the reviewers.

Please take these into account before resubmitting your manuscript:

[LINK]

We look forward to receiving the revised manuscript by Sep 16 2025 11:59PM.   

Sincerely,

Alison Farrell, Ph.D.

Senior Editor 

PLOS Medicine

plosmedicine.org

Requests from Editors:

GENERAL EDITORIAL REQUESTS

* Please use the active voice throughout.

* At this stage, we ask that you include a short, non-technical Author Summary of your research to make findings accessible to a wide audience that includes both scientists and non-scientists. The Author Summary should immediately follow the Abstract in your revised manuscript. This text is subject to editorial change and should be distinct from the scientific abstract. Ideally each sub-heading should contain 2-3 single sentence, concise bullet points containing the most salient points from your study. In the final bullet point of ‘What Do These Findings Mean?’ please include the main limitations of the study in non-technical language.

Please see our author guidelines for more information: https://journals.plos.org/plosmedicine/s/revising-your-manuscript#loc-author-summary.

* Please confirm that your abstract complies with our requirements, including format (three sections: Background, Methods and Findings, and Conclusions) and providing all the information relevant to this study type https://journals.plos.org/plosmedicine/s/submission-guidelines#loc-abstract

* Please ensure that the Introduction ends with a clear description of the study question or hypothesis.

* Please use continuous line numbering. Do not restart numbering on each page.

* Please ensure that where relevant figures include 95% CIs. Please use commas in confidence intervals, throughout.

* Please ensure that all abbreviations are defined at first use throughout the text.

* Please confirm that all numbers presented in the abstract are present and identical to numbers presented in the main manuscript text.

* In the abstract, please include the important dependent variables that are adjusted for in the analyses.

*The Data Availability Statement (DAS) requires revision. For each data source used in your study:

* If new code was created for this analysis, please include the statement on code availability in the data availability statement.

* Please specify whether consent in the relevant studies was informed consent and whether it was written or oral. Please ensure that the research complies with the PLOS policy in full: https://journals.plos.org/plosmedicine/s/human-subjects-research#loc-patient-privacy-and-informed-consent-for-publication

* Please clarify whether this study received approval from a specific IRB. If yes, please identify if in the Methods.

* Figures cannot be reproduced from other sources that are not CC-BY. Please clarify whether this applies to any of the figures, including in SI. Please clarify whether any figures are from the WHO.

* Please define all acronyms used in each figure or table in its corresponding legend.

*Please specify the variables controlled for in all relevant Tables

* In the abstract, please include the important dependent variables that are adjusted for in the analyses.

First sentence in section TDA cascade: phrasing is awkward, please revise.

FUNDING STATEMENT

* The funding statement should include: specific grant numbers, initials of authors who received each award, and URLs (currently missing) to sponsors’ websites. Also, please state whether any sponsors or funders (other than the named authors) played any role in study design, data collection and analysis, the decision to publish, or preparation of the manuscript. If they had no role in the research, include this sentence: “The funders had no role in study design, data collection and analysis, decision to publish, or preparation of the manuscript.”

For Observational studies:

* Please ensure that the study is reported according to the STROBE guideline, and include the completed STROBE checklist as Supporting Information. Please add the following statement, or similar, to the Methods: "This study is reported as per the Strengthening the Reporting of Observational Studies in Epidemiology (STROBE) guideline (S1 Checklist)."

* Did your study have a prospective protocol or analysis plan? Please state this (either way) early in the Methods section.

* For all observational studies, in the manuscript text, please indicate: (1) the specific hypotheses you intended to test, (2) the analytical methods by which you planned to test them, (3) the analyses you actually performed, and (4) when reported analyses differ from those that were planned, transparent explanations for differences that affect the reliability of the study's results. If a reported analysis was performed based on an interesting but unanticipated pattern in the data, please be clear that the analysis was data-driven.

Comments from Reviewers:

Reviewer #1: Thanks to the authors for their answers and the modifications in the manuscript. All my comments have been addressed. I do not have any additional comments.

Reviewer #2: The authors have added explanatory text and substantially improved the manuscript in terms of clarity, accuracy and interpretation.

I have one minor point re. the new sentence (page 14, lines 22-25) "In future, the addition of novel tools with high specificity, such as a three-gene host-response signature previously described to have a high specificity of 90.3% (95% CI 85.5- 94.0) while being moderately sensitive (59.8%, 95% CI 50.8-68.4), might increase overall accuracy and especially specificity."

While I agree with the sentiment that in future, the addition of novel diagnostic tools might increase overall accuracy and especially specificity of the TDAs, I am not sure that reference #19 (Olbrich et al) is accurate for this purpose, since the quoted specificity of 90.3% and sensitivity of 59.8% is for discrimination of Confirmed vs Unlikely TB.

If the authors wish to quote this reference to support use of the 3-gene signature in the TDAs they should quote the specificity of 90.3% and sensitivity of 29.6% for discrimination of Confirmed and Unconfirmed vs Unlikely TB. Alternatively, since sensitivity is sub-optimal, a more generic statement would be appropriate.

Reviewer #3: The paper is clearer and they authors have appropriately addressed the comments. One minor comment is that in Figure 3 on the distribution of scores, for High/Low Risk, the plots are stratified into "Yes" and "No". I would suggest you instead label it as "High Risk" or "Low Risk," which would clarify the plots and would be align with the rest of the figure.

[LINK]

---

## [Editor Report · Decision Letter 3]

25 Sep 2025

Dear Dr van der Zalm, 

On behalf of my colleagues and the Academic Editor, Peter MacPherson, I am pleased to inform you that we have agreed to publish your manuscript "Diagnostic accuracy of the WHO tuberculosis treatment decision algorithms for children with presumptive tuberculosis: an individual participant data meta-analysis" (PMEDICINE-D-25-00818R3) in PLOS Medicine.

PRESS

Sincerely, 

Alison Farrell, Ph.D. 

Senior Editor 

PLOS Medicine